# Vegetation feedback causes delayed ecosystem response to East Asian Summer Monsoon Rainfall during the Holocene

Jun Cheng [1✉], Haibin Wu [2,3,4✉], Zhengyu Liu [5✉], Peng Gu[1], Jingjing Wang[6], Cheng Zhao[6,7], Qin Li[2], Haishan Chen [1], Huayu Lu [6], Haibo Hu [8], Yu Gao[9], Miao Yu[1] & Yaoming Song[1]

One long-standing issue in the paleoclimate records is whether East Asian Summer Monsoon peaked in the early Holocene or mid-Holocene. Here, combining a set of transient earth system model simulations with proxy records, we propose that, over northern China, monsoon rainfall peaked in the early Holocene, while soil moisture and tree cover peaked in the mid-Holocene. The delayed ecosystem (soil moisture and tree cover) response to rainfall is caused by the vegetation response to winter warming and the subsequent feedback with soil moisture. Our study provides a mechanism for reconciling different evolution behaviors of monsoon proxy records; it sheds light on the driving mechanism of the monsoon evolution and monsoon-ecosystem feedback over northern China, with implications to climate changes in other high climate sensitivity regions over the globe.

[1] Key Laboratory of Meteorological Disaster, Ministry of Education (KLME)/Joint International Research Laboratory of Climate and Environment Change (ILCEC)/Collaborative Innovation Center on Forecast and Evaluation of Meteorological Disasters (CIC-FEMD), Nanjing University of Information Science and Technology, Nanjing, China. [2] Key Laboratory of Cenozoic Geology and Environment, Institute of Geology and Geophysics, Chinese Academy of Sciences, Beijing, China. [3] CAS Center for Excellence in Life and Paleoenvironment, Beijing, China. [4] University of Chinese Academy of Sciences, Beijing, China. [5] Atmospheric Science Program, Department of Geography, The Ohio State University, Columbus, OH, USA. [6] School of Geography and Ocean Science, Nanjing University, Nanjing, China. [7] CAS Center for Excellence in Quaternary Science and Global Change, Xian, China. [8] School of Atmospheric Sciences, Nanjing University, Nanjing, China. [9] Laboratory for Climate and Ocean-Atmosphere Studies (LaCOAS) and Department of Atmospheric and Oceanic Sciences, School of Physics, Peking University, Beijing, China. ✉email: chengjun@nuist.edu.cn; haibin-wu@mail.iggcas.ac.cn; liu.7022@osu.edu

A long-standing issue on East Asian Summer Monsoon (EASM) is whether its rainfall peaked in the early Holocene (11.7–8.2 ka) or mid-Holocene (8.2–4.2 ka)[1]. This controversy is clearly apparent over the semi-arid northern China (NC), which is located in the northern margin of the EASM region where the limited rainfall is derived mainly in summer from EASM (Fig. 1a and Supplementary Fig. 1). Regional ecosystem (soil moisture and vegetation types) here is highly sensitive to climate changes in the past[2,3], present[4,5], and future[6], because of the large variation of local EASM rainfall and fragile ecosystem (Fig. 1b). NC is also a key region for the ancient Chinese Civilization[7] and for adaptation to present[5] and future[6] climate changes.

The controversy originates from an apparent inconsistency among proxy records for EASM rainfall. One group of records exhibit an early Holocene peak, which is characterized by a relatively stable evolution from the early to mid-Holocene, followed by a decrease toward the late Holocene (4.2–0 ka[1]); these records include many speleothem isotope records over the broad EASM region[8–10] (Fig. 1c), the [10]Be-based rainfall reconstruction from the Chinese Loess Plateau[11] (Fig. 1d), the lake-level record from Dali Lake[12] (Fig. 1d), and our new annual rainfall reconstruction from fossil pollen in NC for, for example, Daihai Lake (Fig. 1e, red, Supplementary Tables 1 and 2, Supplementary Fig. 2, "Methods," Supplementary Data 1). In contrast, another group of records show a mid-Holocene peak, characterized by a salient wetting trend and increased tree cover from the early to mid-Holocene[2,13–17], followed by a drying trend and declining tree cover afterwards. These records include the pollen-based rainfall reconstruction in Gonghai Lake[13] (blue in Fig. 1e), temperate tree reconstruction[14] (black in Fig. 1f), paleosol[14] (brown in Fig. 1g) and stable dune percentage[15] (cadetblue in Fig. 1g) in NC as well as some speleothem isotope records in southwest China[18]. There are also records that peak between the early and mid-Holocene, such as the recent stalagmite records of Longfeng cave in NC[19].

In spite of decades of studies in various proxy records, this data–data controversy has remained unresolved because of some potential limitations. First, each proxy has its own uncertainty, including its climate interpretation, and this uncertainty is likely to remain in the near future. Second, even for the same proxy, such as the fossil pollen on the same site of Gonghai lake, the reconstructed "climate" signal may differ between different reconstruction methods, for example, between the weighted-averaging partial least squares (WAPLS) calibration function[13] (blue in Fig. 1e) and the Modern Analogue Technique (MAT) (black in Fig. 1e, this study, "Methods"). Third, EASM is a complex monsoon system and its variability is unlikely to be represented by any single index, even at present[20].

This data–data controversy[21] also raises fundamental questions on the evolution mechanism of EASM in the Holocene. The early Holocene peak can be interpreted as a dominant response to summer insolation[3,22,23] (Fig. 2a), while the mid-Holocene peak have been suggested to be caused by a response to the retreating residual Laurentide ice sheet[16,24] and a strengthening Atlantic Meridional Overturning Circulation (AMOC)[13]. Therefore, different responses could imply different dominant forcing mechanisms.

Here, we propose a mechanism for the evolution of NC climate-terrestrial ecosystem in the Holocene by combining proxy records with a set of transient Holocene simulations in a state-of-the-art earth system model ("Methods"). This mechanism suggests a delayed response of the ecosystem to rainfall in NC, offering a solution to reconcile the different proxy evolution behaviors for the EASM there.

## Results

**Paleoclimate–ecosystem simulation.** We first examine the evolution of EASM and the accompanying ecosystem in a transient simulation over the last 21 ka[25] (TraCE21, "Methods"). The simulated EASM monsoon wind, NC rainfall, and the associated water isotopes show a relatively stable or weakly decreasing trend from the early to mid-Holocene, followed by a sharp decline toward late Holocene. This evolution trend is qualitatively consistent with the group of proxies of early Holocene peak (Fig. 1c–e). The coherent evolution of monsoon wind, NC rainfall, and water isotopes has been interpreted as a decreasing EASM moisture convergence and, in turn, rainfall over NC, accompanied by a reduced rainfall upstream over the Indian Ocean and the enriched water isotopes downstream over the broad EASM region[23,24,26].

In contrast to the rainfall peak in the early Holocene, ecosystem variables over NC exhibit a delayed peak in the mid-Holocene. This can be seen in the simulated temperate tree cover (blue in Fig. 1f) and soil moisture (blue in Fig. 1g) over NC, both increasing significantly from the early to mid-Holocene before declining toward the late Holocene. This evolution pattern is consistent with those proxies of mid-Holocene peak[14–17] (Fig. 1f, g, blue in 1e). Although there are some quantitative differences between the model and reconstructions in the evolution of both groups of variables, the delayed response of the mid-Holocene peak to early Holocene peak is clear. The delayed ecosystem response to rainfall suggests a scenario that the hydroclimate and terrestrial ecosystem evolve asynchronously. This asynchronous evolution may offer a solution to reconciling the two groups of proxy records. We note that different hydrological responses across the broad EASM region have been noticed in previous studies[3,27,28] (Supplementary Fig. 4). However, there has been no study on the delayed ecosystem response to rainfall occurring locally in the same region. Understanding this local delayed response should provide a perspective on the climate-ecosystem dynamics of EASM in addition to a solution to this data–data controversy.

**Forcing mechanism on EASM.** The mechanism underlying the evolution of EASM and its delayed ecosystem impact over NC can be understood first by comparing the all forcing experiment of TraCE21 (or ALL) with the accompanying sensitivity experiments under orbital (ORB), ice sheet[29] (ICE), greenhouse gases[30] (GHGs), and meltwater flux (MWF) individually ("Methods"). As expected, the decline of EASM rainfall from its early Holocene peak is forced predominantly by the summer insolation as shown in experiment ORB (red in Fig. 2a). The decreasing summer insolation forces a decrease in the EASM wind (red in Fig. 2b), moisture convergence, and, therefore, rainfall[23,24,28] (red in Fig. 2c) and soil moisture (red in Fig. 2d) over NC throughout the Holocene.

This rainfall decline in NC throughout the Holocene may also be contributed by the enhanced South Asia monsoon via atmospheric teleconnection[23,31], consistent with the enhanced South Asia monsoon in observations[31–33]. The major moisture source to NC appears to originate from the South Asia and Indian Ocean throughout the Holocene (Supplementary Fig. 5a–c). This increased rainfall and, then, moisture from the South Asia can be transported by the mean circulation to the NC region and enhance the rainfall there. Furthermore, NC rainfall seems to be also enhanced by the changing circulation and the associated moisture transport from the western North Pacific from the mid- to early Holocene (Supplementary Fig. 5d) and from the South Asia and Indian Ocean from late to mid-Holocene (Supplementary Fig. 5e). This close relationship between the rainfalls in South Asia and NC, as well as the potential roles of the moisture sources from both the Indian Ocean and the western North Pacific are consistent with a recent study of the Asian monsoon evolution during the last deglaciation[31].

From the early to mid-Holocene, however, this insolation-forced rainfall decline is canceled significantly by the impact of

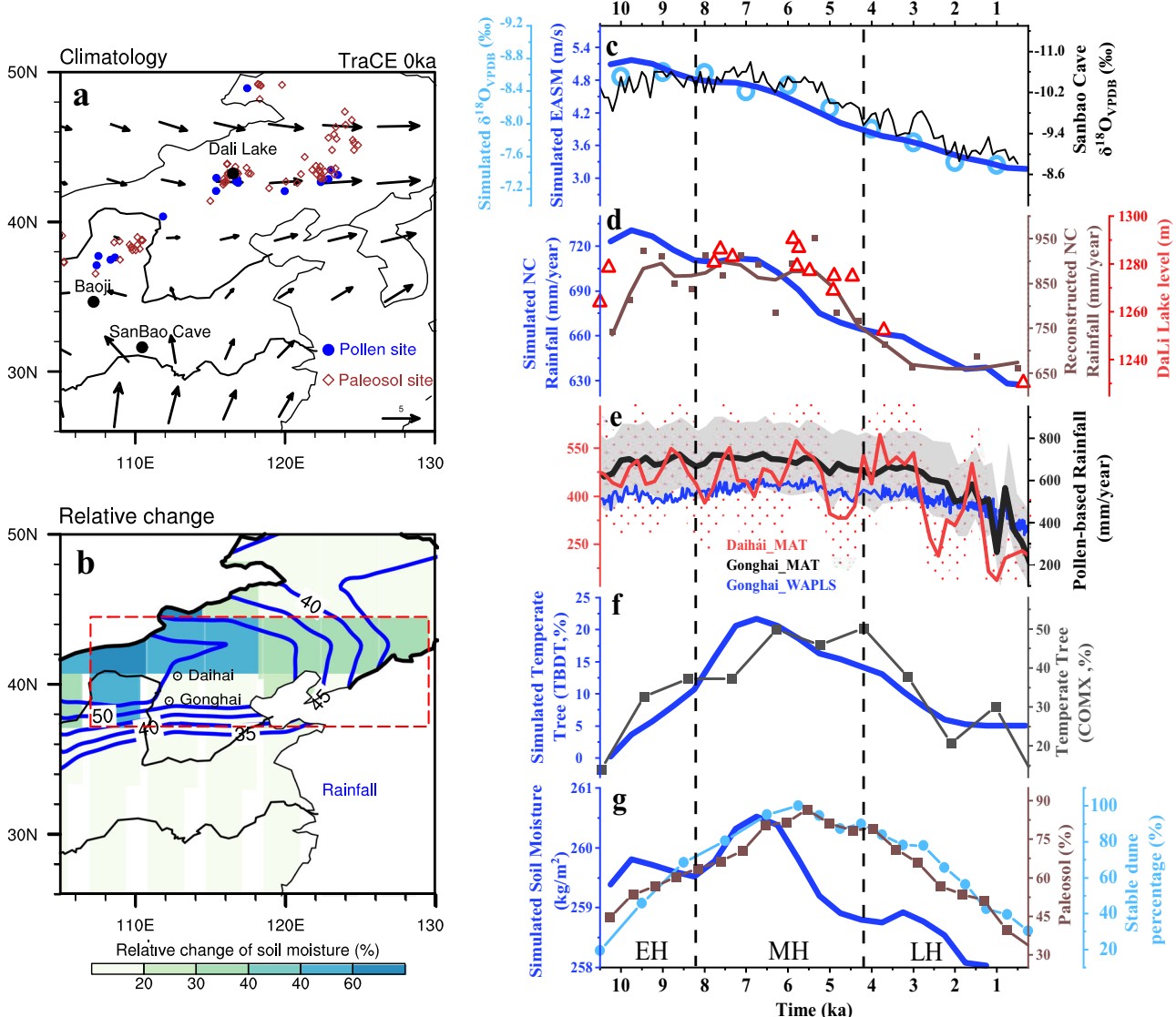

**Fig. 1 Holocene East Asian Summer Monsoon (EASM), Northern China (NC) rainfall and ecosystem in proxy records and TraCE21 simulation. a** Present-day EASM wind in TraCE21 simulation (vector, June to August mean at 850-hPa level). **b** Relative changes (in %) of annual rainfall (blue contours) and upper layer (0.5-m) soil moisture (shading) over Holocene in TraCE21 simulation. **c** EASM evolution as indicated by the 850-hPa meridional wind averaged over the East Asia in TraCE21 simulation (blue), isotope δ[18]O of Sanbao Cave[10] in the proxy record (black curve) and isotope simulations[23] (cadetblue circles). **d** Annual rainfall (blue curve) averaged over the NC region (marked in red-dash box in **b**) in TraCE21 simulation and reconstructed from [10]Be of Baoji on the Chinese Loess Plateau[11] (brown dots and line); lake level of Dali[12] is also plotted in red triangles. **e** Pollen-based reconstruction of annual rainfall in Daihai lake using Modern Analogue Technique (MAT) method (red) and in Gonghai using the methods of MAT (black) and weighted-averaging partial least squares (WAPLS, blue[13]). The 1σ spread (68% confidence interval) of MAT reconstruction is also plotted in dots and shading for Daihai and Gonghai lakes, respectively. **f** Percentage of temperate broad-leave deciduous tree (TBDT) in TraCE21 simulation (blue) and reconstruction from fossil-pollen record of cool mixed tree (COMX[14], black) over NC. **g** Upper layer soil moisture in TraCE21 simulation (blue) and its reconstruction from regional paleosol[14] (brown dots and line) and stable dune percentage over NC[15] (cadetblue dots and line). The sites of records used in **c–g** have been located in **a** and **b**. Temporal spans of the early Holocene (EM), mid-Holocene (MH) and late Holocene (LH) are illustrated with vertical dashed lines in **c–g**.

the retreating ice sheet. The retreat of the residual Laurentide ice sheet from the early to mid-Holocene enhances the EASM rainfall over NC via the circumglobal atmospheric teleconnection[34] and this impact is confirmed in our atmospheric model sensitivity experiments (Supplementary Fig. 6). This enhancement cancels ~2/3 of the insolation-induced rainfall decline over NC (blue in Fig. 2c, f), leaving a weak decreasing trend from the early to mid-Holocene (black in Fig. 2c, f). Relative to the insolation and ice sheet forcing, the impact of MWF on EASM via AMOC is negligible (Supplementary Fig. 7 and Supplementary Discussion).

**Mechanism for the delayed ecosystem response.** The puzzling question here is: over NC, what caused the delayed response in soil moisture and tree cover relative to rainfall? Unexpected to us, one key factor for this delayed response involves vegetation feedback on soil moisture. This can be seen by diagnosing the annual mean hydrological budget for soil moisture. Soil moisture climatology is maintained in general between the moisture source (rainfall) and sinks (bare ground evaporation, plant transpiration, runoff, and internal gravity settling) ("Methods"). Over NC, both the moisture source and sink vary significantly during the Holocene (Fig. 3a, b).

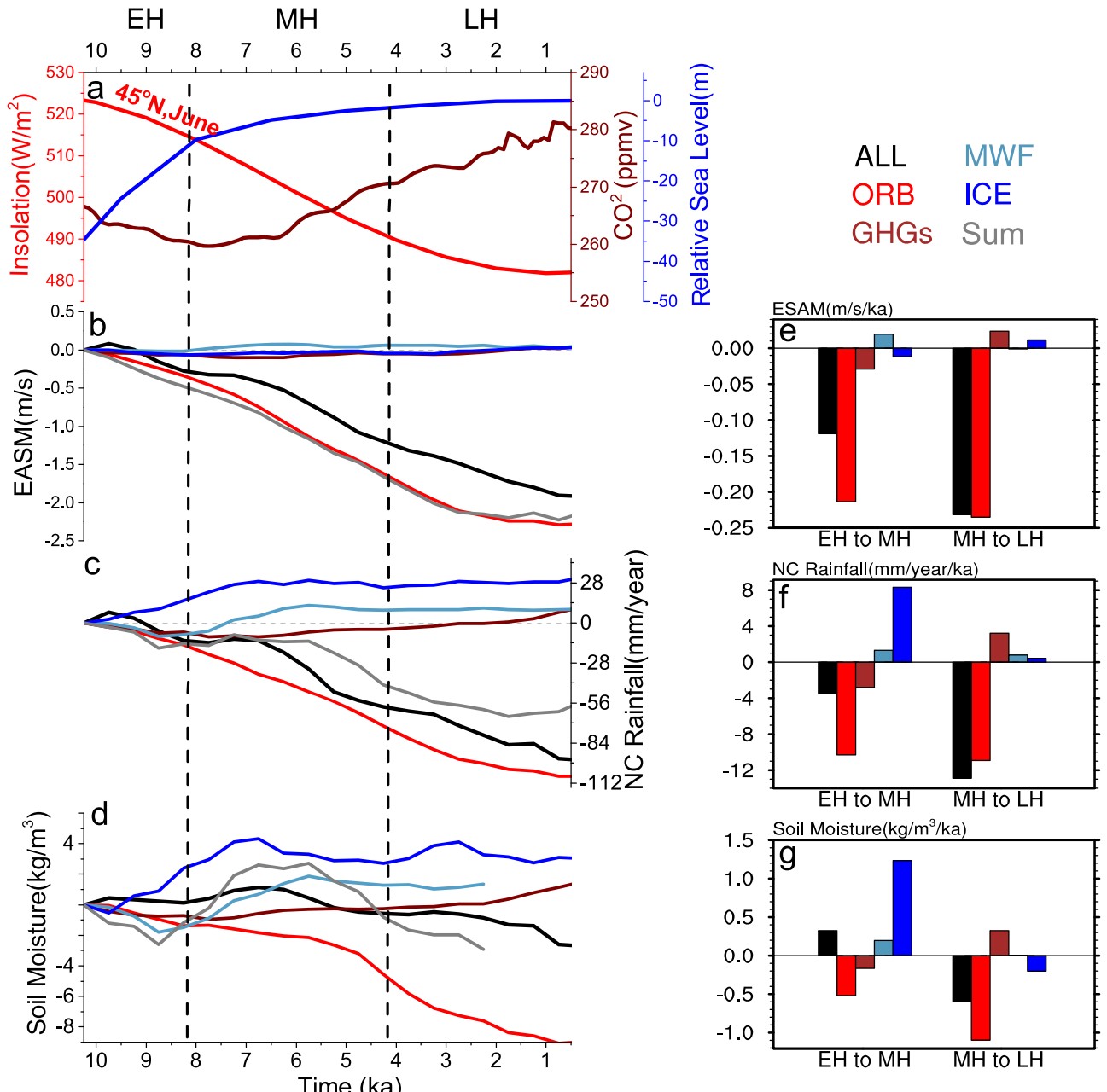

**Fig. 2 East Asian Summer Monsoon (EASM), Northern China (NC) rainfall and soil moisture in TraCE21 simulation and individual forcing experiments.** **a** June insolation at 45°N (red), atmospheric $CO_2$ concentration[30] (brown), and relative sea level[29] (blue) in the Holocene. **b-d** Time evolution of EASM wind, annual mean NC rainfall, and upper layer (0.5-m) soil moisture changes and **e-g** their linear trends for the periods from early to mid-Holocene (EH to MH, 9.5–6.5 ka) and from the mid- to late Holocene (MH to LH, 6.5–0 ka). Shown are the TraCE21 simulation with all forcing (ALL, black), insolation forcing (ORB, red), ice sheet forcing (ICE, blue), greenhouse gases forcing (GHGs, brown), meltwater flux forcing (MWF, cadetblue), and the sum of the four individual forcing experiments (Sum, gray). Temporal spans of the early Holocene (EM), mid-Holocene (MH), and late Holocene (LH) are illustrated with vertical dashed lines in **a-d**.

Most interesting here is the period from the early to mid-Holocene: the weakly decreasing rainfall and increasing ground evaporation (Supplementary Discussion) both favor a drying trend, while other moisture sinks, such as runoff (followed rainfall in semi-arid NC[35]), gravity settling and, in particular, plant transpiration, weaken significantly, favoring a wetting trend (Fig. 3a, b). The wetting trend is so strong that it overwhelms the drying trend, leading to a net moisture gain and soil wetting (Figs. 3c and 1g). After the mid-Holocene, climate evolution is forced by the insolation alone (Fig. 2e–g). The insolation-induced rainfall decline is so strong (tripling the rainfall reduction from the early

to mid-Holocene, black in Fig. 2f) that it overwhelms the change of all the moisture sinks (Fig. 3a, b), leading to a net moisture loss and soil drying (Figs. 3c and 1g).

A further analysis of soil moisture budget points to the reduced plant transpiration as a key factor that generates the soil wetting trend from the early to mid-Holocene (Fig. 3a, b). Transpiration is generated by plant pumping of soil moisture through their roots into air[36] (Fig. 4b). The reduced transpiration from the early to mid-Holocene (Fig. 3a) is caused by a vegetation shift from grasses to trees[37] (Fig. 4a). Since grasses have relatively shallower roots than trees, grasses have a stronger impact on the

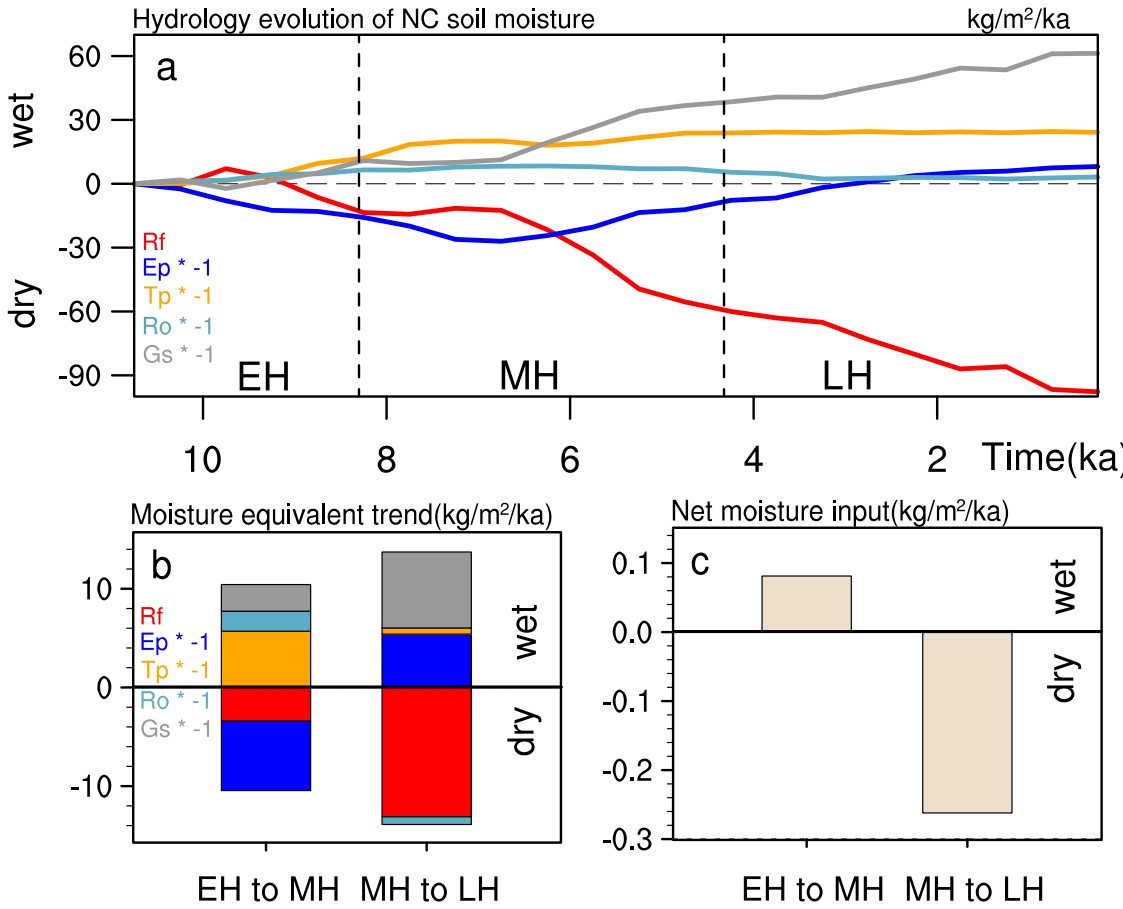

**Fig. 3 Hydrological budget for Northern China (NC) upper layer (0.5-m) soil moisture (SM) in TraCE21 simulation (ALL). a** Time series of each component of SM budget: rainfall (Rf, red), ground evaporation (Ep, blue), plant transpiration (Tp, yellow), surface runoff (Ro, cadetblue), and internal gravity settling (Gs, gray). **b** The moisture-equivalent trend of each hydrological component and **c** the net moisture gain ($\delta SM = Rf - Ep - Tp - Ro - Gs$) of the upper layer soil moisture over the periods of early to mid-Holocene (EH to MH) and of mid-to-late Holocene (MH to LH), respectively. Temporal spans of EM, MH, and LH are illustrated with vertical dashed lines in **a**.

moisture in the upper soil. Thus, decreasing grasses and increasing trees tend to decrease the moisture pumping from the surface soil, which would then reduce the total transpiration (Fig. 4b) and favor surface wetting (Fig. 3b). This vegetation-related transpiration mechanism in the model should be relevant to the real world, because the model vegetation evolution is consistent with the observation of increased tree coverage reconstructed in this region[14] (Fig. 4a).

Ultimately, the vegetation shift and the subsequent surface wetting from the early to mid-Holocene are triggered by the winter warming effect on vegetation. This is demonstrated in further sensitivity experiments with an off-line land-vegetation model ("Methods," Supplementary Table 3), forced by the rainfall and thermal forcing (insolation plus air temperature) in the early and mid-Holocene (Supplementary Table 4). From early to mid-Holocene, the reduced rainfall alone forces a weak drying over NC, but only a small change in vegetation (blue in Fig. 4c, d). In contrast, the thermal forcing alone, characterized by the weak warming in summer and strong warming in winter, shifts grasses to trees and induces strong wetting over NC (red in Fig. 4c, d). Combined, the thermal forcing overwhelms the rainfall effect and forces a wetting response and increased trees (cadetblue in Fig. 4c, d), which are consistent with TraCE21 and the proxy records over NC (Fig. 1f, g). A further experiment with the anomalous thermal forcing in winter only shows that the impact of thermal forcing is caused almost completely by the winter warming (brown in Fig. 4c, d). These analyses suggest that the delayed vegetation

response in the early to mid-Holocene is caused ultimately by the winter warming, which raises the coldest temperature above the threshold ($-17\,°C$)[36] for the survival of broadleaf trees and the extinction of grasses (Fig. 4a) and favor a shift from grasses to trees. It should be noted that, since wetter soil also favors trees over grasses[35], the vegetation shift and soil wetting triggered by winter warming might be further enhanced by the positive feedback with soil moisture in a fully coupled climate-ecosystem such as that in TraCE21.

## Discussion

Our study proposes a mechanism for a delayed response of terrestrial ecosystem to rainfall in the Holocene. The declining summer insolation reduces the monsoon rainfall. The increasing winter insolation, however, favors trees over grasses and, in turn, a surface wetting due to the interaction with vegetation and soil moisture. This wetting, aided by the retreat of the residual ice sheet, overwhelms the drying due to summer insolation from the early to mid-Holocene (Fig. 2c), generating a net soil wetting and ecosystem response with a ~3000–4000 years delay to rainfall. This mechanism offers a potential resolution to the data–data controversy on the peak timing of EASM in the NC region: the proxies of early Holocene peak may register more signature of local rainfall, while the proxies of mid-Holocene peak may represent more the signal of terrestrial ecosystem.

Our study has further implications to the role of vegetation feedback on climate not only for the past, but also for the present

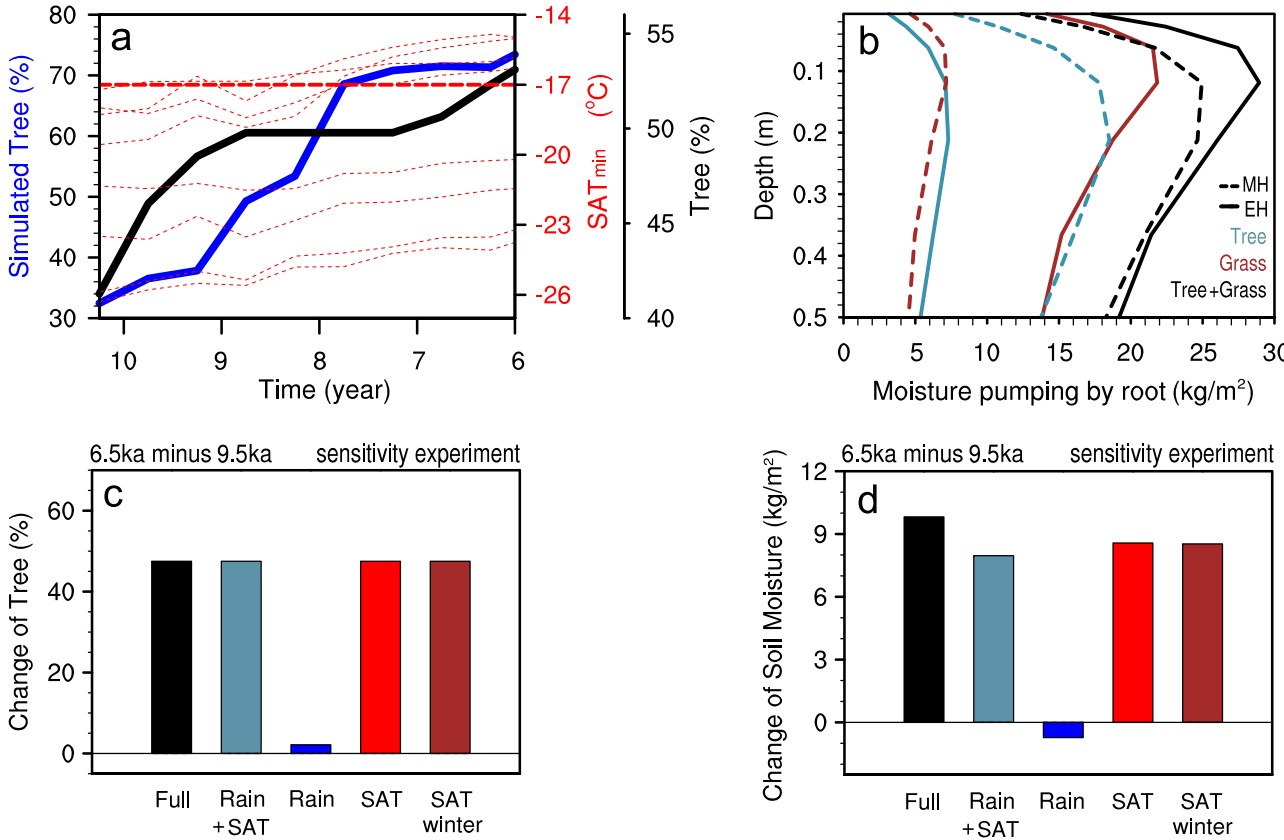

**Fig. 4 Change of tree percentage and moisture pumping over Northern China (NC) from the early Holocene (EH) to mid-Holocene (MH). a** Evolutions of tree percentage in proxy records[14] (black) and TraCE21 simulation (blue) over the period of EH to MH. Coldest temperature in the winter season ($SAT_{min}$) in each grid box within the NC region is also shown as thin red dashed lines, with the temperature threshold (−17 °C) for broadleaf trees and grass in the model as the thick red-dash line. **b** Vertical profiles of the moisture pumping by root for trees (cadetblue), grasses (brown), and the sum (black) in EH (9.5 ka, solid) and MH (6.5 ka, dash). Changes of **c** tree percentage and **d** soil moisture from EH to MH in an off-line land-vegetation-coupled model that forced by TraCE21 atmospheric variables in different combinations: full forcing (Full, black), rainfall and temperature forcing (Rain + SAT, cadetblue), rainfall only (Rain, blue), temperature only (SAT, red), and winter temperature only (SAT winter, brown).

and future. It has potential implications to the tree-planting campaign currently implemented over NC[5], calling for further studies of vegetation feedback on hydroclimate and ecosystem. The tree-planting could invoke a positive vegetation feedback in relatively humid regions, but a negative vegetation feedback in relatively arid regions. Moreover, our study has global implications on other monsoon regions[38] and semi-arid areas of high climate sensitivity, such as northern Africa and Australia, where the ecosystem response may also lag the rainfall in the Holocene, as shown in TraCE21 (Supplementary Fig. 8), even though the exact mechanisms may differ. Our study may also shed light on a question on the driving mechanism of the monsoon evolution[39]: Is the monsoon intensity in phase[22] or not in phase[40] with the precessional forcing? Part of the controversies among different proxies could be caused by their different representations of different variables in the climate ecosystem[41–43].

## Methods

**Pollen-based reconstruction of NC rainfall using MAT**. Our analysis is based on the climatic interpretation of fossil-pollen stratigraphy from lakes of Gonghai and Daihai in NC (black and red in Fig.1e, Supplementary Fig. 2, and Supplementary Table 1). These two fossil-pollen datasets are among a group of pollen records that meet the following criteria: (1) the raw pollen spectra should consist of representative pollen types and be well-suited for documenting the natural vegetation changes given the relatively limited human activities impact, (2) the records should have a reliable chronology with a minimum of seven independent $^{14}C$ age control points, (3) the records should have a resolution of better than 200 years, and (4) the records should contain the core top samples, such that they span the time window from the past to

the present without depositional hiatuses. The age-depth models were updated and calibrated using Clam 2.2[44] in R code. All ages in the text refer to calendar years.

We used a modern pollen dataset as the basis for analogue match to the fossil data. The modern pollen data were derived from the East Asian Pollen Database including 1756 pollen spectra[45], the China Quaternary Pollen Database including 465 pollen spectra[46], with which we updated 380 new pollen spectra. After excluding repetition sites and those with obvious human impact, a total of 1865 pollen spectra were included in this present study (Supplementary Fig. 2), which cover most regions of China.

Modern climate data including the mean annual temperature and mean annual precipitation at the site of each modern sample were derived from WorldClim climate data version 2[47] via extraction by R code. WorldClim 2 dataset is interpolated at 1-km grid using monthly station climate data (1970–2000) with covariates including elevation, distance to the coast, and MODIS-derived maximum and minimum land surface temperature and cloud cover, which improve the interpolation accuracy particularly for temperature variables at the extreme elevations.

Paleoclimate reconstruction used the MAT in this study (black and red in Fig. 1e)[48–52], and was carried out by the R package "Rioja"[53]. We directly match fossil samples with modern datasets through their dissimilarity of biome score compositions, measured by the squared-chord distance method (Supplementary Fig. 3). Because biomes, groups of dominant plants characterized by common phenological and climate constraints, can reduce the poor analogue situations, anthropogenic/non-climatic influence, and thus reflect climate more stably[50,51,54]. Scores are obtained in the biomization procedure followed by China Quaternary Pollen Database[46].

A dissimilarity threshold (T = 0.06) for non-analog/analog definition is applied to decrease the risk of non-analog bias (Supplementary Discussion). Then the dissimilarity-weighted mean climate of at most seven closest modern analogues was assigned to the fossil sample[55,56]. We used linear interpolation for each site to produce a 200-yr-step sequence after the paleoclimate reconstruction.

**Statistical analyses**. MAT model performance is evaluated by the root mean square error and the coefficient of determination ($R^2$) based on leave-one-out

cross-validation using the modern datasets. The results show that the predictions closely match the observations, with a better performance for precipitation than temperature (Supplementary Table 2), indicating a higher sensitivity to precipitation change. Full errors during our reconstruction procedure are considered, accounting of three main sources: (1) errors in modern biome-climate calibration and application for each sample (sample-specific errors) and (2) temporal interpolation errors for each site. For each source, Monte Carlo or bootstrap resampling approach is used to assess the uncertainty. The 1σ spread (68% confidence limit) to the reconstructed climate curve is given by combining all the errors in Fig. 1e.

**Comparison of Gonghai Lake reconstructions in two methods**. Our Gonghai Lake reconstruction using MAT (Fig. 1e, black) is based on the similarity between modern analogues and fossil-pollen samples. It searches for the $k = 7$ closest modern assemblages for each fossil sample using squared-chord distances, and calculates the reconstructed paleoclimate ANNP as the weighted mean of the modern sample sites[48,49,52] (Supplementary Discussion). In comparison, the ANNP reconstruction from Chen et al.[13] (Fig. 1e, blue) uses the WAPLS. This method is based on unimodal relationships between pollen percentages and climate, and the pollen percentages were transformed to square-roots to optimize the "signal" to "noise" ratio. In addition, it uses the additional PLS components to diminish bias and to increase performance[13,57,58]. The two reconstructions are roughly consistent with each other, with the annual rainfall amount decreasing from the peak of ~600 mm/year in the early to mid-Holocene to the minimum of ~300 mm/year at the present. A careful comparison, however, shows a shift of the mid-Holocene peak in WAPLS toward the early Holocene in MAT, making the latter like an early-Holocene peak (blue to black in Fig. 1e), although the WAPLS reconstruction is still within the 1σ spread of the MAT reconstruction. This difference of the peak timing between the two methods is an indication of the potential uncertainty of the rainfall reconstruction from Gonghai lake.

**TraCE21 transient climate-terrestrial ecosystem simulations**. TraCE21 is a transient simulation of the global climate-terrestrial ecosystem evolution in the last 21,000 years in the fully coupled model of CCSM3[59], with the spatial resolution of T31 (3.75° × 3.75°[57]) in the atmosphere. Experiment TraCE21 (or Exp. All) is forced by the realistic climatic forcing of orbital insolation[60], atmospheric GHGs[30], meltwater discharge, and the continental ice sheets (ICE-5G[61]). The coastlines and bathymetry are changed at 13.1, 12.9, 7.6, and 6.2 ka for the Barents Sea, the Bering Strait, Hudson Bay, and the Indonesian through-flow, respectively[25,62]. This simulation reproduces a reasonable climate evolution from the LGM to the Holocene[63,64], and the major climate events such as H1, BA, and YD[24,65–68]. On regional scales, the experiment seems to be able to produce a reasonably simulation for the present climatology (Supplementary Fig. 1) and long-term change of the EASM[23,24].

In addition to Experiment TraCE21 (or All) that is forced by all the forcing factors, four sensitivity experiments are performed. Each sensitivity experiment is forced individually by one of the four time-evolving forcing factors of orbital forcing (Exp. ORB), continental ice sheet (Exp. ICE), greenhouse gases (Exp. GHGs), and meltwater fluxes (Exp. MWF), while other forcing factors are fixed at their 19-ka value[62,66].

To focus on the variation on orbital scale, the model output was first binned in 500-year length and then smoothed with a 3-point running mean. In the simulation, the NC region is defined as the area of 108°–129°E, 37°–45°N as shown in Fig. 1b.

The time spans for early and mid-Holocene are defined as 11.7–8.2 kaBP and 8.2–4.2 ka by IUGS[1]. In TraCE21 simulation, we chose the trend between 9.5 and 6.5 ka to represent the changes from early to mid-Holocene, and that between 6.5 and 0 ka for the changes from mid- to late Holocene, which could reasonably indicate the recorded responding changes.

**EASM index in TraCE21**. EASM index is defined as the area-averaged summer (June to August, JJA) meridional wind in East China (110°–120°E, 27°–37°N) at 850-hPa level, following Liu et al.[23].

**Soil moisture in model simulations**. The upper layer soil moisture here is calculated as the averaged soil moisture of the top 0.5 m in the land models.

**Hydrological budget analyses in soil**. Hydrological balance equation of soil moisture can be written as

$$\rho H\, \delta SM_i = \left(Rf_i - Ep_i - Tp_i - Ro_i - Gs_i\right)\delta t \tag{1}$$

where $\rho(= 1000\,kg/m^3)$ is the density of soil moisture, $H$ (=0.5 m) is the thickness of the upper layer soil, $\delta SM_i$ is the change of soil moisture over model integration interval of $\delta t$, $Rf_i$, $Ep_i$, $Tp_i$, $Ro_i$, and $Gs_i$ are, respectively, rainfall, evaporation, plant transpiration, runoff, and internal gravity settling at the depth $H$, with $i$ indicating the time step of model integration. Based on the vertical root profile of each plant type and their composition ratio, moisture pumping at depths is

calculated for each plant as the sum in the upper 0.5-m layer as the plant transpiration $Tp$. The gravity settling $Gs$ is then calculated at each time step from the equation above as the residual.

**Sensitivity experiments in the land-vegetation-coupled model**. To clarify the mechanism of the plant transition during Holocene shown in the proxy records and the TraCE21 simulation, we performed a series of sensitivity experiments of land-vegetation coupling using the Community Land Model version 4 in the NCAR CESM1.2.0 at a single point locating in NC (40.18°N, 116.25°E, Supplementary Table 3). We first forced the model with the solar radiation flux at surface (Insolation), rainfall, surface air temperature (SAT), surface wind, specific humidity (Q), and surface pressure (PS) of TraCE21 (Supplementary Table 4) to perform a pair of experiments at 9.5 ka (Exp.1) and 6.5 ka (Exp.2) with the same initial state (including soil moisture and plants) that are derived from the 9.5 ka in TraCE21. The difference of these two control experiments (Exp.2–Exp.1, black in Fig. 4c, d) reproduced the changes of tree percentage and soil moisture in proxy records and TraCE21 (Fig. 1e, f).

These sensitivity experiments (Supplementary Table 3) show that the surface wetting over NC is caused predominantly by the winter warming on tree growth. For convenience, the impact of insolation and SAT are considered together as the thermal forcing in the sensitivity experiments here. (The impact of insolation alone is much smaller than that of SAT, because the impact of insolation on land and vegetation has been incorporated mainly into the effect of SAT in the model.) Based on the control experiment of 9.5 ka (Exp.1), the sensitivity experiments are designed to quantify the impact of rainfall (Exp.4–Exp.1), thermal forcing (Exp.5–Exp.1), the combined effect of rainfall and thermal forcing (Exp.3–Exp.1), and winter thermal forcing (Exp.6–Exp.1). These sensitivity experiments integrated for 210 years with the average of the last 50 years used for analysis.

The responding amplitude of vegetation and soil moisture in these experiments is almost twice that in TraCE21 from early to mid-Holocene. This difference likely caused by the different land-ecosystem model from that in TraCE21. However, qualitatively, the model produces the same results as in TraCE21 and therefore can be used to shed light on the mechanism of the ecosystem response in TraCE21 (For comparison with the results of TraCE21, we scaled the values in Fig. 4c, d with a factor of 2).

## Data availability

The TraCE21 simulation outputs used in this study are available on the Earth System Grid at the NCAR (https://www.earthsystemgrid.org/dataset/ucar.cgd.ccsm3.trace.html). Source data are provided with this paper.

## Code availability

The codes that support the findings of this study are available upon request from the corresponding authors.

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

## Acknowledgements

This work is supported by the Strategic Priority Research Program of the Chinese Academy of Sciences (XDA13010106), Chinese MOST2016YFA0600504, NSFC41630527, NSFC 41888101, MOST 2017YFA0603801, and US NSF P2C2.

## Author contributions

J.C., Z.L., and H.W. conceived the study; H.W. and Q.L. performed the reconstruction of Northern China rainfall in pollen; P.G., J.W., H.H., Y.G., M.Y., and Y.S. performed the analysis and experiments; C.Z., H.C., and H.L. contributed to the interpretation of proxy records and ecosystem; Z.L., J.C., and H.W. wrote the paper with inputs from all authors.

## Competing interests

The authors declare no competing interests.
