## [Peer Review File · Nature Communications]

REVIEWER COMMENTS

Reviewer #1 (Remarks to the Author):

The East Asian summer monsoon (EASM) is very important climate system in northern hemisphere, as its behaviors strongly affect the eco-environment and human welfares in this heavy populated area. While, the EASM system dynamics seems quite complicated, because it is not only related to low latitudes and high latitudes earth-climate system, but also is connected to ocean and continental interior processes. Especially, vegetation ecology is very sensitive to precipitation and temperature variations in this region. The relationship between vegetation in northern China (NC) and EASM evolution is an interesting topic. This study suggested that the vegetation feedback in early Holocene caused delayed response of high precipitation in NC based on earth system model experiments and proxy records. This manuscript provide some 'Novel' idea to of early Holocene high precipitation, low exportation and low soil moisture. However, the high precipitation in early Holocene don't have solid evidence in EASM area, because this area was largely influenced by the lower sea level and longer distance of water vapor transportation from the Pacific in the early Holocene. The modeling result of lower evaporation in the early Holocene needs more evidence, and the importance of winter isolation on vegetation might have been exaggerated as many studies revealed early Holocene forest well development in colder area than NC. In general, the major conclusions of this manuscript might be misleading and need more persuasive evidence.

Here below are some major comments.

1. The Quaternary term of 'early Holocene' may have different time span from that of the climate modeling study. The authors should state the time span of early and middle Holocene in their study at the beginning, and the possible discrepancies of climate histories based on modeling and geological proxies.
2. In Line 34-35, in the summarized two groups of evidence, authors stated that the speleothem isotope records support 'early Holocene peak' with stable climate, while, recent new study compiled several Holocene speleothem sequence, which suggested that early Holocene EASM was not stable and the stable 'high peak' appeared in 8-6 ka of mid-Holocene (Yang Xunlin et al., 2019 The Holocene).
3. The methodology of MAT is totally based on the similarity between the pollen assemblages of fossil and modern samples. The problem of this method is that the climate settings in the early Holocene, as this study mainly concerned, should be totally different from those of nowadays, and the fossil pollen percentage has large possibility of no-analogue in modern pollen dataset, combining the effect of strong human impacts on the modern vegetations. Therefore, based on MAT, the reconstructed early Holocene climate is not convincing enough. The original pollen based reconstruction of precipitation in Daihai Lake area was very low in early Holocene (Xu Q.H. et al. 2010 Journal of Climate).
4. The another 'early Holocene peak' evidence provided by this manuscript is the lake level evolution of Dali Lake, while, the reconstructed lake level is not continuous and has been questioned as well (Goldsmith Y. and Xu H., 2020 JQS).
5. The big logic problem of the modeling results is that the precipitation was high, and the evaporation was low, but the soil moisture was low in the early Holocene!!! Although the authors gave some explanations about this, it is still lack of solid evidence to support the suggested mechanism. In the semi-arid area, numerous studies suggested that the higher precipitation will enhance the above-ground biomass in annual to decadal timescale (e.g. Bai Y et al., 2004 Nature), and then higher precipitation would have caused soil formation on the Loess Plateau in early Holocene. However, with the synthesis investigated Holocene loess-paleosol on the loess plateau in NC, the early Holocene paleosol was not so developed as that in the middle-Holocene (Wang H. et al., 2014 QI), which also indicated lower precipitation.

6. One of the main statement of this manuscript is that winter insolation shaped the vegetation in early to middle Holocene. This conclusion seems not correct. As many palynological studies reconstructed fast vegetation development in early Holocene over northern Europe, central Siberia and northeastern China, where have colder winter than northern China. For example, in Sihailongwan Marr Lake, the temperature species was high in early Holocene (Stebich et al., 2015 QSR).

Reviewer #2 (Remarks to the Author):

The authors tackle the long-standing debate over the timing of the maximum intensity of the EASM in the Holocene. They use new pollen records and climate model simulations to investigate the response of rainfall and ecosystem in northern China, which they chose because of its sensitivity to changes in rainfall, given that it is on the northern fringe of the EASM domain. They find that the discrepancies between proxy records can be reconciled when both precipitation and ecosystem processes (tree cover and soil moisture) are taken into account, which results in a later peak of ecosystem response than the rainfall amount (which is principally driven by orbital forcing).

There are a few things that the authors need to address/change before I can recommend the manuscript for publication.

Most importantly, I think the manuscript lacks clarity on what the specific issues are that are addressed. The issue that the authors describe (different peaks in EASM reconstructions) appears to be generated by different sensitivities of the proxies and archives used in the reconstructions. To my understanding, the reconstructions showing a later "peak precipitation" thus would not be registering precipitation but rather the ecosystem response, which in this case is somehow decoupled from precipitation. This is now not clearly explained, and therefore there was some confusion for me as to what exactly the different proxies show. I think this could be addressed if the authors provided a few sentences at the end discussing these issues, and this would make the message of the paper much clearer.

Moreover, I find the term "eco-environment" used throughout the manuscript to be ambiguous. It is not clear what part of the ecosystem is responding, until the discussion, where the authors specifically only refer to vegetation type and soil moisture. This needs to be clarified earlier on.

My second point is that, while the authors discuss the shortcomings of proxy records and their climatic signal in detail, there is no such discussion for the model simulations. However, models are not infallible, and it is important to recognize potential biases that could result from these simulations. A discussion reflecting these issues is needed to balance the manuscript.

Thirdly, until I read the methods section it was not clear to me that the authors had produced one of the pollen records for this study! A few sentences in the main manuscript on the methods used (both proxy records and models) would improve the readability and understanding of the reader.

Technical comments:

Line 26: "peaking of the EASM" is very ambiguous language. What is meant here: a peak of rainfall intensity, peak in seasonal distribution,... ? I suggest clarifying this term throughout the manuscript.

Lines 79 and following: I don't understand the reasoning for having this paragraph. These are clearly two very different things, one relating to regional differences in precipitation, the other one to a decoupling of precipitation and ecosystem response. I suggest deleting it, because it adds nothing to the discussion.

Results and discussion sections: please add the timings defined for the Early Holocene and Mid Holocene time periods.

Line 96: the impact of the ice sheet is to my understanding only in NC, please clarify.

Figure 1: A – I think it's confusing that some sites (Dali Lake, Sanbao Cave, Baoji) are shown by a larger dot and labelled, whereas other record locations are barely distinguishable. I suggest using a different colour than red for the dots and clarifying in the caption why those locations are highlighted.

B – what does the red box signify?

C and D – please clarify what the difference between EASM and annual rainfall in NC is.

Figure 2: Please specify what the dashed line in A-D is.

Reviewer #3 (Remarks to the Author):

Review of "Vegetation Feedback Leads to Delayed Eco-Environmental Response to East Asian Summer Monsoon Rainfall during Holocene"

The authors present a study of the changing environmental conditions over Northern China from the early to mid-Holocene. This is used to examine the resulting impacts on ecosystems and to help understand the discrepancy in timing between moisture variations and tree cover. I thought this was a very neat study and a great example of the combined use of proxy data and model output to tackle an existing question. The background to the study was well described, and the study setup allows the changes over this region to be explored from large spatial scales (moisture transport) to small scales (soil moisture). I found the results to be well presented and clearly support the arguments that are made. I have a few minor comments but otherwise am happy to recommend this for publication.

- My main comment concerns the last paragraph of the discussion. This lists several broader outcomes of the study, including feedbacks from current tree planting, or longer-term changes in monsoon forcing. However, these are simply raised as questions, with no attempt to answer them. While I understand that to fully answer these will require significant further work, the authors could extrapolate from their current findings to make some suggestions, to help reinforce the potential applications of their work

- Line 16: I'm not sure what is meant here by "The evolution behavior..."

- Line 113: In this line, declines in runoff are taken as evidence of wetting. I would have expected reduced runoff to result from drier conditions. Please expand on this section.

- Line 124: The difference in evapotranspiration between grasslands and forests has been demonstrated previously. The authors might want to cite Yamazaki et al (2004, J. Hydrometeor.) for example.

- Line 206: How was the threshold of 0.06 chosen? Or the choice of 7 analogs?

Reviewer #1 (Remarks to the Author):

The East Asian summer monsoon (EASM) is very important climate system in northern hemisphere, as its behaviors strongly affect the eco-environment and human welfares in this heavy populated area. While, the EASM system dynamics seems quite complicated, because it is not only related to low latitudes and high latitudes earth-climate system, but also is connected to ocean and continental interior processes. Especially, vegetation ecology is very sensitive to precipitation and temperature variations in this region. The relationship between vegetation in northern China (NC) and EASM evolution is an interesting topic. This study suggested that the vegetation feedback in early Holocene caused delayed response of high precipitation in NC based on earth system model experiments and proxy records. This manuscript provides some 'Novel' idea to of early Holocene high precipitation, low exportation and low soil moisture. However, the high precipitation in early Holocene don't have solid evidence in EASM area, because this area was largely influenced by the lower sea level and longer distance of water vapor transportation from the Pacific in the early Holocene. The modeling result of lower evaporation in the early Holocene needs more evidence, and the importance of winter isolation on vegetation might have been exaggerated as many studies revealed early Holocene forest well development in colder area than NC. In general, the major conclusions of this manuscript might be misleading and need more persuasive evidence.

Re: Thanks for your detailed review and constructive comments.

1) As you mentioned, temperature is an important factor controlling the vegetation variation in northern China (NC), its robust impact has been clearly illustrated in our manuscript, which could even dominate the wetting of NC from early to mid- Holocene through the hydrological impact of vegetation variation as we suggested.

2) We agree that there is no solid evidence for high precipitation in early Holocene in geological records. If there were, this would no longer be a problem. Instead, in observations, some evidences show mid-Holocene peak while some evidences not. Mechanistically, it is understandable that the signal of precipitation variation can be blended with that of other factors (e.g. temperature) in different ways in various geological records. This data-data inconsistency is indeed our original motivation of this study. The major contribution of our paper is to provide a plausible (novel) physical mechanism that enables the precipitation of early Holocene peak to co-evolve with some eco-environment variables of mid-Holocene peak, and therefore offer one plausible solution to the controversy among proxies. Our experiments also imply, from the modeling perspective, the pure signal of the precipitation variation is hard to extract precisely from the eco-environmental records. In Fig.1, we intend to show evidences on two sides, those of mid-Holocene peak (Fig.1F&G) and those of early Holocene peak, or at least no significant increase trend from early to mid-Holocene (Fig.1D&E) as well as some other proxies (Fig.1C). We also illustrated the potential uncertainty of proxy reconstruction of precipitation: the reconstructed precipitation highly relied on the reconstruction method with fossil pollen, the mid-Holocene peak of the reconstructed precipitation based on the Weighted Averaging-Partial Least Squares (WAPLS) in Lake Gonghai (Chen et al., 2015) becomes less significant if we reconstructed it in another scheme using the Modern Analogue Technique (MAT) (Fig.1E).

3) The impact of sea level changes has been considered in TraCE21 simulation (Methods 4). The simulated pattern of water vapor transportation to NC wasn't changed in the early Holocene relative to mid-Holocene (Fig.E1a&b), because the sea level changes were less than 40 m (Siddall et al., 2003; Liu et al., 2004) and thus did not significantly effect on the land area extension. Our model simulation indicates that the water vapor transported (and in turn precipitation) is larger in the early Holocene because of the stronger monsoon circulation that is forced by the orbital forcing in the early Holocene (Fig.E1c). In general, the land area effect with sea level change has been speculated for a long time. Yet, to our knowledge, there is no modeling studies that suggest that the land area effect is of critical importance during the Holocene for NC precipitation. Instead, it is the circulation change that played the dominant role.

Fig. E1. Vertical integrated vapor transport from surface to 500 hPa at early Holocene (EH, a), mid-Holocene (MH, b) and their difference (EH-MH, c). Brown line indicates the northern fringe of EASM in present day.

4) Modern investigations for the hydrological processes in semi-arid regions indicates that evaporation is constrained by soil moisture (Seneviratne et al., 2010), which implies that the evaporation in the early Holocene should be low in response to the recorded low soil moisture in NC; this is confirmed in our model simulation.

5) One point should be clarified, the temperature ranges are dramatically different for the survival of different types of vegetation (Bonan et al., 2003). For example, the coldest minimum air temperature is about -17°C for the survival of temperate broadleaf deciduous tree in NC, but is much lower (e.g. -32.5°C) for that of boreal needleleaf evergreen tree in colder regions than NC. We will reply this point later with more details. Therefore, the importance of winter insolation on

vegetation does not seem to be exaggerated in our model.

Ref:

F. Chen, Q. Xu, J. Chen, H. J. B. Birks, J. Liu, S. Zhang, L. Jin, C. An, R. J. Telford, X. Cao, Z. Wang, X. Zhang, K. Selvaraj, H. Lu, Y. C. Li, Z. Zheng, H. Wang, A. Zhou, G. Dong, J. Zhang, X. Huang, J. Bloemendal, Z. Rao, East Asian summer monsoon precipitation variability since the last deglaciation. *Sci. Rep.* 5, 11186 (2015).

J.P. Liu, J.D. Milliman, S. Gao, P. Cheng, Holocene development of the Yellow River's subaqueous delta, North Yellow Sea. *Mar. Geol.* 209, 45–67 (2004).

M. Siddall, E.J. Rohling, A. Almogi-Labin, C. Hemleben, D. Meischner, I. Schmelzer, D.A. Smeed, Sea-level fluctuations during the last glacial cycle. *Nature* 423, 853–858 (2003).

S. I. Seneviratne, T. Corti, E. L. Davin, M. Hirschi, E. B. Jaeger, I. Lehner, B. Orlowsky, A. J. Teuling, Investigating soil moisture-climate interactions in a changing climate: A review. *Earth-Sci. Rev.* 99, 125-161 (2010).

G. B. Bonan, S. Levis, S. Sitch, M. Vertenstein, K. W. Oleson, A dynamic global vegetation model for use with climate models: concepts and description of simulated vegetation dynamics. *Glob. Chang. Biol.* 9, 1543-1566 (2003).

Here below are some major comments.

1. The Quaternary term of ‘early Holocene’ may have different time span from that of the climate modeling study. The authors should state the time span of early and middle Holocene in their study at the beginning, and the possible discrepancies of climate histories based on modeling and geological proxies.

Re: Thanks for the suggestions. We have stated the time spans for early, mid- and late Holocene defined by IUGS 2018 (Walker et al., 2018) in the main text as “the early Holocene (11.7-8.2 ka) or mid-Holocene (8.2-4.2 ka)” (Lines 28). We also stated the time spans used in modeling study in Methods part and their consistency within modeling and geological proxies as “In TraCE21, we chose the trend between 9.5 and 6.5 ka to represent the changes from early to mid-Holocene, and that between 6.5 and 0 ka for the changes from mid- to late Holocene, which could reasonably indicate the recorded responding changes.” (Lines 272-275) and figure captions (Lines 521-522).

Ref:

M. Walker, M.J. Head, M. Berkelhammer, S. Björck, H. Cheng, L. Cwynar, D. Fisher, V. Gkinis, A. Long, J. Lowe, R. Newnham, S. Rasmussen, H. Weiss, Formal ratification of the subdivision of the Holocene Series/Epoch (Quaternary System/Period): two new Global Boundary Stratotype Sections and Points (GSSPs) and three new stages/subseries. *Episodes*[online]: [https://doi.org/10.18814/epiugs/2018/018016\(2018\)](https://doi.org/10.18814/epiugs/2018/018016(2018)).

2. In Line 34-35, in the summarized two groups of evidence, authors stated that the speleothem isotope records support ‘early Holocene peak’ with stable climate, while, recent new study compiled several Holocene speleothem sequence, which suggested that early Holocene EASM was not stable and the stable ‘high peak’ appeared in 8-6 ka of mid-Holocene (Yang Xunlin et al., 2019 The Holocene).

Re: Thank you for the suggested reference, which is indeed interesting. We think these records do not affect our major conclusion. First, we present one speleothem isotope record that is representative of, at least, one group of speleothem records as one evidence of early Holocene peak. The revised text states “these records include **some** speleothem isotope records over the broad EASM region” (Line 38), instead of **all** isotope records. Second, we also included other existing reconstruction (lake level) as well as a new reconstruction of precipitation reconstructions as evidences suggesting the “early Holocene peak”.

We did now, include the new reference (18), to highlight the potential uncertainties even within speleothem records over China. Since the relation between those speleothem records and NC precipitation is uncertain, these records themselves, either early Holocene peak or mid-Holocene peak, do not provide the definite answer of the precipitation changes in NC. The different speleothem evolution may reflect the asynchronous variation of precipitation over broad EASM regions as discussed in our manuscript (Lines 85-86). The speleothem isotope record with “early Holocene peak” also has been reproduced in an isotope simulation whose EASM peaked in early Holocene (Liu et al., QSR2014).

Ref:

Z. Liu, X. Wen, E. C. Brady, B. Otto-Bliesner, G. Yu, H. Y. Lu, H. Cheng, Y. Wang, W. Zheng, Y. Ding, R. L. Edwards, J. Cheng, W. Liu, H. Yang, Chinese cave records and the East Asia Summer Monsoon. *Quat. Sci. Rev.* 83, 115-128 (2014).

3. The methodology of MAT is totally based on the similarity between the pollen assemblages of fossil and modern samples. The problem of this method is that the climate settings in the early Holocene, as this study mainly concerned, should be totally different from those of nowadays, and the fossil pollen percentage has large possibility of no-analogue in modern pollen dataset, combining the effect of strong human impacts on the modern vegetations. Therefore, based on MAT, the reconstructed early Holocene climate is not convincing enough. The original pollen based reconstruction of precipitation in Daihai Lake area was very low in early Holocene (Xu Q.H. et al. 2010 *Journal of Climate*).

Re: Thanks for your comments. The MAT introduction and its distinguishing feature are presented in method section on lines 192-235 in revised manuscript.

The paleoclimate in our study is reconstructed using the modern analogue technique (MAT) (Refs: 44-45), due to its outstanding utility for continental/regional reconstruction (Refs: 46-48). Modern analogues of fossil sample are the closest surface samples measured by squared chord distance (SCD). Then the dissimilarity-weighted mean climate of the 7 closest modern analogues is assigned to fossil sample for paleoclimate reconstruction.

In this study, biome score from the Plant Functional Type (PFT), instead of direct pollen assemblages, are used in the dissimilarity calculation. Because PFTs, groups of dominant plants characterized by common phenological and climate constraints, can reduce the poor analogue situations, anthropogenic/non-climatic influence during the early Holocene and thus reflect climate more stably and accurately (Refs: 46-47, 50).

The modern pollen data were derived from the East Asian Pollen Database (EAPD) including

1756 pollen spectra (Ref: 41), the China Quaternary Pollen Database including 465 pollen spectra (Ref: 42), with which we updated 380 new pollen spectra. Although human activities may impact on the modern vegetation, samples in which the pollen assemblage is dominated by agricultural crops and/or weeds as these represent anthropogenic disturbance were excluded in this study. A total of 1865 pollen spectra representing the potential nature vegetation were included (Fig. S2), which cover most regions of China with wide ranges of climate changes, more than the modern pollen samples (N=237) of Xu et al. (2010) only cover in the northern China.

MAT model performs quite well in annual precipitation reconstruction, with higher coefficient of determination ($R^2=0.91$) based on leave-one-out cross-validation in this study than that ($R^2=0.86-0.88$) in Xu et al. (2010). At the same time, our relatively low and stable SCD with below 0.02 throughout the Holocene, between fossil samples and modern analogues (Fig.E2) indicates a general good analogous situation for all the Holocene, further ensuring this reconstruction reliability.

Fig. E2 Squared chord distance between fossil samples and modern analogues during the Holocene for Daihai Lake

Although we present the MAT reconstruction not as the truth, instead, we present it to show the potential uncertainty in the different reconstruction of precipitation, even on the same pollen records. As you stated above, the signal of precipitation variation could be blended by that of temperature in the fossil pollen and is hard to be extracted precisely with current methodologies. As a result, its reconstructions using different methodologies could be different (Fig. 1E). That's why it's critically important for us to cross-check among different geological records and to combine them together with model simulation to further understand the possible changes of EASM precipitation in NC over Holocene period.

4. The another 'early Holocene peak' evidence provided by this manuscript is the lake level evolution of Dali Lake, while, the reconstructed lake level is not continuous and has been questioned as well (Goldsmith Y. and Xu H., 2020 JQS).

Re: Thanks for your constructive comments. We have corrected the lake level of Lake Dali during the Holocene in Fig.1D. Although it shows the uncertainty of the paleo-lake level reconstructions on the same lake Dali, as for Daihai and speleothem discussed above. Our view is that this

controversy can't be resolved with the same proxy alone, or even with all the proxies. It is critically important to combine proxies with modeling to understand the mechanism. Our paper is one step in this direction.

5. The big logic problem of the modeling results is that the precipitation was high, and the evaporation was low, but the soil moisture was low in the early Holocene!!! Although the authors gave some explanations about this, it is still lack of solid evidence to support the suggested mechanism. In the semi-arid area, numerous studies suggested that the higher precipitation will enhance the above-ground biomass in annual to decadal timescale (e.g. Bai Y et al., 2004 Nature), and then higher precipitation would have caused soil formation on the Loess Plateau in early Holocene. However, with the synthesis investigated Holocene loess-paleosol on the loess plateau in NC, the early Holocene paleosol was not so developed as that in the middle-Holocene (Wang H. et al., 2014 QI), which also indicated lower precipitation.

Re: Thanks for your comments. We agree that, usually, soil moisture tend to follow precipitation. This is true when the precipitation trend is strong such that any soil/vegetation feedback will be overwhelmed by the forcing. This is indeed the case in our model after mid-Holocene when both precipitation and soil moisture decrease. Nevertheless, our proposed mechanism can still be correct for several reasons.

First, in the period from early to mid-Holocene, precipitation decrease is not substantial in the model because of the competing impact by orbital and ice sheet. As such, other factors could become important to “tip” the subtle balance and thus lead to weakly opposite trends between soil moisture and precipitation. In this case, in our model simulation, the dominant factor that controls the vegetation changes from grass to temperate broad-leave deciduous forest could be the winter warming, as further assessed in our sensitive experiments with land-vegetation coupled model.

Second, it should also be pointed out that the relation between precipitation and vegetation may vary depending on time scales. The present-day observed relationship between precipitation and vegetation in NC may not work for the millennial evolution from early to mid-Holocene, because of the different dominating factors.

Third, there are uncertainties in proxy records on paleosol as in precipitation. The paleosol changes on the Loess Plateau is consistent with our presented evidence of paleosol records (Ref: 14) and stable dune percentage records (Ref: 15) in NC, which all indicate the wetting trend from early to mid-Holocene and this trend is simulated by the model. As shown in our experiments, the precipitation changes from early to mid-Holocene is not the key factor controlling the wetting trend and then the enhanced pedogenesis of paleosol in NC.

Fourth, present-day studies of hydrological processes in semi-arid regions indicate that evaporation can be constrained by the soil moisture (Seneviratne et al., 2010), implying a lower evaporation in the early Holocene in response to the recorded low soil moisture in NC and this seems to be consistent with our model simulation.

In general, given the relatively subtle moisture balance in this period of early to mid-Holocene, the uncertainty with various records, potential feedbacks among different components of the land-atmosphere systems, our proposed mechanism, and the relationship in the land-atmosphere system, can be potentially correct.

6. One of the main statement of this manuscript is that winter insolation shaped the vegetation in early to middle Holocene. This conclusion seems not correct. As many palynological studies reconstructed fast vegetation development in early Holocene over northern Europe, central Siberia and northeastern China, where have colder winter than northern China. For example, in Sihailongwan Marr Lake, the temperature species was high in early Holocene (Stebich et al., 2015 QSR).

Re: Thanks for your comments. First, we pointed out that this winter temperature control mechanism proposed here is particularly effective in the particular period (early to mid-Holocene) and region (NC). This does not apply to any region in any time. Second, the fast development of vegetation in early Holocene in these regions you mentioned were simulated too, indicating the reasonable simulating ability of TraCE21 for the evolution of global vegetation. In fact, past evolutions of vegetation were complex because they were controlled by many factors besides temperature, as you mentioned above. Nevertheless, in our model simulation, the dominant factor that controls the NC vegetation changes from early to mid-Holocene could be the winter warming, as further assessed in our sensitive experiments with land-vegetation coupled model.

In colder regions of northern Europe and central Siberia than NC, the vegetation types (e.g., boreal summergreen tree, boreal evergreen conifer) are different with that in NC (e.g., temperate summergreen tree, cool-temperate conifer) and their survival coldest minimum air temperature is lower (e.g. $\leq -32.5^{\circ}\text{C}$, Bonan et al., 2003), which could lead to an earlier fast development in early Holocene when precipitation was not the limit factor (Fig.E3). The dominant vegetation types were temperate deciduous forest (TEDE) and cool mixed forest (COMX) during the Holocene in Sihailongwan Marr Lake. Its COMX increased from early to mid-Holocene, which is similar with that in NC but the increase amplitude is relative small (Stebich et al., 2015 QSR). COMX is the dominant vegetation type in NC (Ref: 14) and the coldest minimum air temperature for its survival is higher (around -17°C). From early to mid-Holocene, the winter insolation (may also involving the decreasing of ice sheet) increased the coldest minimum air temperature from below to up the threshold value and then induced the significant response of COMX.

Fig. E3 Simulated changes in vegetation type and associated winter temperature in TraCE21 for periods before early Holocene (a-c) and from early to mid-Holocene (d-f). Locations of winter temperature contours for -17°C (red) and -32.5°C (purple) are illustrated for periods of 15ka (dotted line), 10.5ka (dashed line) and 6.5ka (solid line).

Reviewer #2 (Remarks to the Author):

The authors tackle the long-standing debate over the timing of the maximum intensity of the EASM in the Holocene. They use new pollen records and climate model simulations to investigate the response of rainfall and ecosystem in northern China, which they chose because of its sensitivity to changes in rainfall, given that it is on the northern fringe of the EASM domain.

They find that the discrepancies between proxy records can be reconciled when both precipitation and ecosystem processes (tree cover and soil moisture) are taken into account, which results in a later peak of ecosystem response than the rainfall amount (which is principally driven by orbital forcing).

Re: Thanks for your detailed review and constructive summaries for our manuscript.

There are a few things that the authors need to address/change before I can recommend the manuscript for publication.

Most importantly, I think the manuscript lacks clarity on what the specific issues are that are addressed. The issue that the authors describe (different peaks in EASM reconstructions) appears to be generated by different sensitivities of the proxies and archives used in the reconstructions. To my understanding, the reconstructions showing a later “peak precipitation” thus would not be registering precipitation but rather the ecosystem response, which in this case is somehow decoupled from precipitation. This is now not clearly explained, and therefore there was some confusion for me as to what exactly the different proxies show. I think this could be addressed if the authors provided a few sentences at the end discussing these issues, and this would make the message of the paper much clearer.

Re: Thanks for your constructive suggestion. We have rewritten our discussion section to highlight the major issues addressed in the paper. We have also written the “here, we...” statement (Line 60-64) to highlight these major issues as follows: “Here, we propose a new mechanism for the evolution of NC climate-terrestrial ecosystem in the Holocene by combining proxy records with a set of transient Holocene simulations in a state-of-the-art earth system model. This mechanism suggests a delayed response of the ecosystem to rainfall in NC, offering a new resolution for reconciling the different proxy evolution behaviors for the EASM there.” Essentially, our study is motivated by the issue of data-data inconsistency on the timing EASM during the Holocene. We first proposed a new mechanism and then use this mechanism to offer a resolution to the data-data inconsistency. We hope the revision is more clear.

Moreover, I find the term “eco-environment” used throughout the manuscript to be ambiguous. It is not clear what part of the ecosystem is responding, until the discussion, where the authors specifically only refer to vegetation type and soil moisture. This needs to be clarified earlier on.

Re: Thanks. We replaced the “eco-environment” with “ecosystem” throughout the manuscript and specified it in abstract (Lines 7-9) as “we propose that, over northern China, EASM rainfall peaked in the early Holocene, while soil moisture and tree cover peaked in the mid-Holocene. The delayed ecosystem (soil moisture and tree cover) response to rainfall ...” and in main text (Line 31) as “Regional ecosystem (soil moisture and vegetation types)”.

My second point is that, while the authors discuss the shortcomings of proxy records and their climatic signal in detail, there is no such discussion for the model simulations. However, models are not infallible, and it is important to recognize potential biases that could result from these simulations. A discussion reflecting these issues is needed to balance the manuscript.

Re: Thanks. The discussion about the model biases has been added in Lines 93-99 as “The amplitude of the change of tree cover (Fig. 1F) and soil moisture (Fig. 1G) in the model are smaller than those in the corresponding proxy records, and the simulated area of NC response (Fig. 1B) also seems smaller than that indicated by proxy records (Fig. 1A). These differences may be caused by the model deficiency, including its coarse resolution, as well as proxy uncertainties. In spite of the potential model biases, the consistency between the evolution phase in the records and simulation still could provide the basis to investigate the underlying mechanism for the delayed ecosystem response to NC rainfall.”.

Thirdly, until I read the methods section it was not clear to me that the authors had produced one of the pollen records for this study! A few sentences in the main manuscript on the methods used (both proxy records and models) would improve the readability and understanding of the reader.

Re: Thanks. We have clarified it in Lines 40-41 as “our new annual rainfall reconstructions from fossil pollen in NC for, for example, Daihai Lake”.

Technical comments:

Line 26: “peaking of the EASM” is very ambiguous language. What is meant here: a peak of rainfall intensity, peak in seasonal distribution,... ? I suggest clarifying this term throughout the manuscript.

Re: Thanks for your suggestion and we have clarified these. As we discussed in the manuscript, EASM itself is a complex climate system and involves many components, including wind and rainfall, and may not be the same within the entire EASM region. As such, there is no unique way to represent the EASM intensity using a single index. In our manuscript, for large scale EASM, we use one traditional representation related to the wind in eastern China. For local climate in NC, we use rainfall to represent the intensity. This local representation in rainfall is more directly related to the impact on ecosystem. We have revised the relevant expressions in main text. For example, we added in Lines 71-74: “Here, the monsoon wind refers to the southerly wind averaged in eastern China, as a representation of large scale moisture transport of the EASM system (23), while the NC rainfall is a regional EASM feature that is more related to the local proxy records.”

Lines 79 and following: I don't understand the reasoning for having this paragraph. These are clearly two very different things, one relating to regional differences in precipitation, the other one

to a decoupling of precipitation and ecosystem response. I suggest deleting it, because it adds nothing to the discussion.

Re: Thanks for your comments. We have revised this part in Lines 85-89 to focus on our main concerns as “We note that different hydrological responses across the broad EASM region have been noticed in previous studies (3, 26, 27, Fig. S3). However, there has been no study on the delayed ecosystem response to rainfall occurring locally in the same region. Understanding this local delayed response should provide a new perspective on the climate-ecosystem dynamics of EASM and may further offer a resolution to the data-data controversy.”.

Results and discussion sections: please add the timings defined for the Early Holocene and Mid Holocene time periods.

Re: The time span of early and mid-Holocene has been clarified in Line 28 as “the early Holocene (11.7-8.2 ka) or mid-Holocene (8.2-4.2 ka)”. The calculation scheme for the trend between these two periods has been clarified in Lines 272-275 as “The time spans for early and mid-Holocene are defined as 11.7-8.2 ka and 8.2-4.2 ka by IUGS (1). In TraCE21, we chose the trend between 9.5 and 6.5 ka to represent the changes from early to mid-Holocene, and that between 6.5 and 0 ka for the changes from mid- to late Holocene, which could reasonably indicate the recorded responding changes.”. Thanks for your suggestion.

Line 96: the impact of the ice sheet is to my understanding only in NC, please clarify.

Re: Clarified in Line 112 as “with a minor impact on the southern regions of EASM”. Thanks.

Figure 1: A – I think it’s confusing that some sites (Dali Lake, Sanbao Cave, Baoji) are shown by a larger dot and labelled, whereas other record locations are barely distinguishable. I suggest using a different colour than red for the dots and clarifying in the caption why those locations are highlighted.

Re: The labelled sites in Fig. 1A indicate the records used in Fig. 1C&D, the unlabeled dots indicate the records used in the synthesized tree percentage in Fig. 1F and paleosol percentage in Fig.1G. The labelled sites have been changed to black. The relevant expressions in the caption have been revised.

B – what does the red box signify?

Re: The area used in the NC rainfall in D, relevant expression has been revised in Lines 502-503.

C and D – please clarify what the difference between EASM and annual rainfall in NC is.

Re: The relationship between ESM and NC rainfall has been clarified in Lines 71-77 as “Here, the monsoon wind refers to the southerly wind averaged in eastern China, as a representation of large scale moisture transport of the EASM system (23), while the NC rainfall is a regional EASM feature that is more related to the local proxy records. The coherent evolution of monsoon wind, NC rainfall and water isotopes has been interpreted as a decreasing EASM moisture convergence and, in turn, rainfall over NC, accompanied by a reduced rainfall upstream over the Indian Ocean and the enriched water isotopes downstream over the broad EASM region (22, 25).” As indicated in Fig. E4, the changes of annual rainfall in NC follows its summer part, indicating the dominance of EASM over this region.

Fig. E4 NC averaged rainfall in annual (red) and summer (JJA, blue) over Holocene period.

Figure 2: Please specify what the dashed line in A-D is.

Re: Vertical dashed line has been modified to two to illustrate the corresponding period of early, mid- and late Holocene in Fig. 1, 2 and 3. It's also specified in Lines 514, 524-525 in figure captions.

Reviewer #3 (Remarks to the Author):

Review of “Vegetation Feedback Leads to Delayed Eco-Environmental Response to East Asian Summer Monsoon Rainfall during Holocene”

The authors present a study of the changing environmental conditions over Northern China from the early to mid-Holocene. This is used to examine the resulting impacts on ecosystems and to help understand the discrepancy in timing between moisture variations and tree cover. I thought this was a very neat study and a great example of the combined use of proxy data and model output to tackle an existing question. The background to the study was well described, and the study setup allows the changes over this region to be explored from large spatial scales (moisture transport) to small scales (soil moisture). I found the results to be well presented and clearly support the arguments that are made. I have a few minor comments but otherwise am happy to recommend this for publication.

Re: Thanks for your detailed review and constructive comments.

- My main comment concerns the last paragraph of the discussion. This lists several broader outcomes of the study, including feedbacks from current tree planting, or longer-term changes in monsoon forcing. However, these are simply raised as questions, with no attempt to answer them. While I understand that to fully answer these will require significant further work, the authors could extrapolate from their current findings to make some suggestions, to help reinforce the potential applications of their work.

Re: We have made some extrapolations at the end of the paper in a new single paragraph.

- Line 16: I'm not sure what is meant here by “The evolution behavior...”

Re: Revised in Line 17 as “How did East Asian Summer Monsoon (EASM) rainfall vary over Holocene”.

- Line 113: In this line, declines in runoff are taken as evidence of wetting. I would have expected reduced runoff to result from drier conditions. Please expand on this section.

Re: In semi-arid NC, the decline of runoff followed the decrease of rainfall, the condition of soil moisture is not a key factor to determine the changes of runoff. We have expanded this point in Lines 127-128 as “runoff (followed rainfall in semi-arid NC, 31)”.

- Line 124: The difference in evapotranspiration between grasslands and forests has been demonstrated previously. The authors might want to cite Yamazaki et al (2004, J. Hydrometeor.) for example.

Re: Thanks for your recommendation and the reference has been cited as ref 33 in Line 138.

- Line 206: How was the threshold of 0.06 chosen? Or the choice of 7 analogs?

Re: Thanks for your comments.

Paleoclimate reconstruction will be more reliable if the fossil sample has close modern analogues in the training-set. To find a suitable dissimilarity threshold (T), there is a trade-off between the reconstruction precision/accuracy and the utility for the majority of samples. T value (< 0.06) was determined by the relatively low root mean squared error of prediction (RMSEP), high coefficient of determination (R^2) and over 98% of surface samples having analogues (Fig. E5), through conducting a series of cross-validation experiments following Williams and Shuman (2008).

Fig.E5 RMSEP, R^2 and percentage of samples with analogues changes along with dissimilarity threshold for annual precipitation calibration.

Paleoclimate is reconstructed using the modern analogue technique (MAT), modern analogues of fossil samples are the closest surface samples measured by squared chord distance (SCD), and then in general the dissimilarity-weighted mean climate of the 3-7 closest modern analogues is assigned to fossil sample (Williams and Shuman, 2008). In this study, we chose the 7 closest modern analogues (Fig. E6) based on the relationship between the number of modern analogues and the changes of R^2 , RMSEP to maximize the precision and accuracy for paleoclimate reconstruction.

Fig.E6 R² and RMSEP with the changes of number of modern analogues for annual precipitation calibration.

Ref:

J. W. Williams, B. Shuman. Obtaining accurate and precise environmental reconstructions from the modern analog technique and North American surface pollen dataset. *Quaternary Sci. Rev.* 27, 669-687 (2008).

REVIEWER COMMENTS

Reviewer #1 (Remarks to the Author):

Thanks for your detailed explanations and answers for my questions.

I believe that this MS provides some plausible explanations for discrepancies of different proxy records mainly based on modelling work.

I only have one question now:

It is clearly shown in Fig. E1. 'Vertical integrated vapor transport from surface to 500 hPa at early Holocene ...' in the response file that the dominant water vapor in NC area was contributed by Indian Summer Monsoon (ISM) in the early and middle Holocene, and the East Asia Summer Monsoon (EASM) system contributed little water vapor. And, many studies in ISM area have shown that the ISM was much stronger in early Holocene than that in the middle and late Holocene (e.g. Xiao XY et al., 2020 QSR Dianchi Lake pollen records; Wu Duo et al., 2019 QSR isotope records: Decoupled early Holocene summer temperature and monsoon precipitation in southwest China; and other more records from stalagmites in southern China and India). If these reconstructions were right, the stronger ISM could contribute more water vapor to NC in the early Holocene than middle Holocene. But the figure 1E c doesn't show this.

So my question is why the NC area high precipitation in early Holocene was NOT caused by stronger ISM but EASM (Fig. E1 c)? I am confused with this.

Reviewer #2 (Remarks to the Author):

I thank the authors for the thorough revisions they provided on their manuscript. This is a neat study combining paleoclimate data and modelling and presenting a new and intriguing possible explanation for some long-standing controversy in the Asian monsoon region.

I think the changes made are sufficient and have strengthened the manuscript, and I am happy to recommend it for publication in Nature Communications.

The only thing I would recommend is to check the wording of the manuscript, as some parts could benefit from clarification and there also were still some typos.

Reviewer #3 (Remarks to the Author):

I thank the authors for their detailed responses to my previous comments, and the improvements they have made on the manuscript. In particular, I think that this has improved a) the context that frames their research; b) the description of the interaction between precipitation and evaporation trends and c) the potential implications of the study.

My only remaining comment is that it would be useful to include some of the material in the authors' responses in the SI (e.g. the plot of SCD over time, and the description of how they chose the number of analogs and the threshold). Other than that, I am happy to recommend this for publication.

Reviewer #1 (Remarks to the Author):

Thanks for your detailed explanations and answers for my questions.

I believe that this MS provides some plausible explanations for discrepancies of different proxy records mainly based on modelling work.

I only have one question now:

It is clearly shown in Fig. E1. 'Vertical integrated vapor transport from surface to 500 hPa at early Holocene ...' in the response file that the dominant water vapor in NC area was contributed by Indian Summer Monsoon (ISM) in the early and middle Holocene, and the East Asia Summer Monsoon (EASM) system contributed little water vapor. And, many studies in ISM area have shown that the ISM was much stronger in early Holocene than that in the middle and late Holocene (e.g. Xiao XY et al., 2020 QSR Dianchi Lake pollen records; Wu Duo et al., 2019 QSR isotope records: Decoupled early Holocene summer temperature and monsoon precipitation in southwest China; and other more records from stalagmites in southern China and India). If these reconstructions were right, the stronger ISM could contribute more water vapor to NC in the early Holocene than middle Holocene. But the figure 1E c doesn't show this.

So my question is why the **NC area high precipitation in early Holocene was NOT caused by stronger ISM but EASM** (Fig. E1 c)? I am confused with this.

Re: Thanks for your further comments.

First, TraCE21 did simulate a stronger ISM in rainfall, monsoon wind, and vapor transport, in the early Holocene (Fig. R1d), which is consistent with the records you suggested. This ISM response coexists with our NC climate-ecosystem response in TRACE21 and therefore is dynamically consistent in the model. This is consistent with Fig.R1b (Fig. E1c in Response 1) in a stronger convergence of moisture transport into both the ISM region and NC. This enhanced monsoon in both ISM and NC, however, are all forced by the external forcing, mainly insolation, as shown explicitly in our single forcing experiments.

Second, the simultaneously stronger ISM and NC may be related to a teleconnection from ISM to NC, as discussed in our previous work (ref. 22). This teleconnection generates perturbation circulation that can enhance moisture into NC via enhanced velocity on mean moisture ($v'Q_{\text{mean}}$), consistent with Fig. R1b. It does not, however, necessarily mean the enhanced NC rainfall has to derive all of its additional moisture from the ISM region, which would require a more detailed Lagrangian study, which is well beyond the scope of this work.

Third, regardless if NC is enhanced by ISM or not, our major point here is the delayed response of soil moisture from rainfall. Nevertheless, we think this is an interesting point that can be added in the discussion of the mechanism. Therefore, we have incorporated this point and the new references in the paper as follows (Lines 231-234):

“This early Holocene rainfall peak in NC may also be enhanced partly by the enhanced South Asia monsoon via atmospheric teleconnection in our model (22), consistent with the enhanced South Asia monsoon in observations (30, 31).”

Fig. R1. Vertical integrated vapor transport from surface to 500 hPa at early Holocene (EH, a), its change from early to mid-Holocene (EH-MH, b) and that from mid- to late Holocene (MH-LH, c). d, South Asia area-averaged rainfall (blue line, 20°-30°N, 85°-95°E) and vertical integrated vapor transport (green line) and zonal wind (red line) from surface to 500 hPa (10°-30°N, 60°-100°E).

Reviewer #2 (Remarks to the Author):

I thank the authors for the thorough revisions they provided on their manuscript. This is a neat study combining paleoclimate data and modelling and presenting a new and intriguing possible explanation for

some long-standing controversy in the Asian monsoon region.

I think the changes made are sufficient and have strengthened the manuscript, and I am happy to recommend it for publication in Nature Communications.

The only thing I would recommend is to check the wording of the manuscript, as some parts could benefit from clarification and there also were still some typos.

Re: Thanks for your further comments and recommendation.

We have also made significant effort in checking and improving the English.

Reviewer #3 (Remarks to the Author):

I thank the authors for their detailed responses to my previous comments, and the improvements they have made on the manuscript. In particular, I think that this has improved a) the context that frames their research; b) the description of the interaction between precipitation and evaporation trends and c) the potential implications of the study.

My only remaining comment is that it would be useful to include some of the material in the authors' responses in the SI (e.g. the plot of SCD over time, and the description of how they chose the number of analogs and the threshold). Other than that, I am happy to recommend this for publication.

Re: Thanks for your further comments and recommendation.

We have moved the plot of SCD over time and the description of choosing analogs number and threshold to the SI.

REVIEWER COMMENTS

Reviewer #1 (Remarks to the Author):

Thanks for your reply.

The modelling results (Fig R1-b) are still confusing.

Here is the simple logic:

- 1) the ISM is mainly driven by summer insolation.
- 2) the ISM is dominant water vapor source of NC area.
- 3) the insolation and ISM were stronger in the early Holocene (EH) than that of the middle Holocene (MH).
- 4) The water vapor in NC contributed by ISM (WVNCISM) should be $EH > MH$.

But, the Fig R1-b doesn't show this.

And it shows $(WVNCISM)_{EH} - (WVNCISM)_{MH} = \text{zero (nearly)}$, which is quite strange and more explanations are necessary.

And the modelling result becomes normal again in the Fig.R1-C, $(WVNCISM)_{MH} - (WVNCISM)_{LH} > 0$.

In my opinion, the slightly higher precipitation in NC_EH (if it is true) could be more likely contributed by stronger ISM, and some recent study in NC also stated this (Tan L et al., 2020 GRL).

And on the other hand, new evidence supports that 9.5-5 ka was the wettest time period in NC suggested by stalagmite records of Longfeng cave in NC (Wei et al., 2020 GPC), and stalagmite growth should have no delayed response as ecosystem (as authors suggested). So, the modelling results of 9.5 ka and 6.5 ka (Line 434) could not represent the whole EH (11.7-8.2ka) and MH (8.2-4.2 ka) respectively, as both of the two time-slides are located in the wettest period.

Reviewer #1 (Remarks to the Author):

We thank reviewer for the careful review and new references.

The reviewer has raised two concerns. The first is on the relation between ISM and NC rainfall, and the second is on the relation between rainfall and a new moisture record over NC. We will address these two questions separately.

The modelling results (Fig R1-b) are still confusing.

Here is the simple logic:

- 1) the ISM is mainly driven by summer insolation.
- 2) the ISM is dominant water vapor source of NC area.
- 3) the insolation and ISM were stronger in the early Holocene (EH) than that of the middle Holocene (MH).
- 4) The water vapor in NC contributed by ISM (WVNCISM) should be $EH > MH$.

But, the Fig R1-b doesn't show this.

And it shows $(WVNCISM)_{EH} - (WVNCISM)_{MH} = \text{zero (nearly)}$, which is quite strange and more explanations are necessary.

And the modelling result becomes normal again in the Fig.R1-C, $(WVNCISM)_{MH} - (WVNCISM)_{LH} > 0$.

In my opinion, the slightly higher precipitation in NC_EH (if it is true) could be more likely contributed by stronger ISM, and some recent study in NC also stated this (Tan L et al., 2020 GRL).

Re: The question here is the relation between ISM and NC rainfall. We will discuss this from several angles.

1. The relation between ISM and NC rainfall.

In the “logic” above. “1”, “2”, and “3 are correct, 4 may not be true.

First, we clarify the relation between the change in rainfall and total moisture transport vq and the composition of the latter. The increase of rainfall in a region (such as NC) should be

associated with an increase in the total moisture transport νq (more precisely, its convergence, please see later discussion). This total moisture transport response however can be decomposed into two components: the mean (i.e. climatological) transport $\bar{\nu}$ on the perturbation (i.e. response) moisture Δq (i.e. $\bar{\nu} \Delta q$) and the perturbation transport $\Delta \nu$ on mean moisture \bar{q} (i.e. $\bar{q} \Delta \nu$).

$$\Delta(\nu q) \approx \bar{\nu} \Delta q + \bar{q} \Delta \nu$$

The first term $\bar{\nu} \Delta q$ can be seen roughly from Fig. R1a and its pattern depends mainly on the climatological mean wind (multiplied by q). The strong wind $\bar{\nu}$ out of the ISM region suggests that if the moisture is increased over ISM $\Delta q > 0$ (associated with increased rainfall), this extra moisture does cause an increase of moisture exported downstream. This will enhance the total moisture export out of ISM. This is true for both periods EH-MH and MH-LH, because both cases have the similar climatological $\bar{\nu}$ and increased $\Delta q > 0$ over ISM region.

The second term $\bar{q} \Delta \nu$ is reflected more in Fig.R1b and c, and its pattern depends critically on the perturbation (response) of circulation $\Delta \nu$. Therefore, “(WVNCISM)EH - (WVNCISM)MH = zero (nearly)” judged from Fig.R1b suggests that from MH to EH, the contribution of perturbation circulation export of moisture from ISM is small. The change of total moisture export out of ISM can still be enhanced by the mean transport term $\bar{\nu} \Delta q$. For the MH to EH, both terms contribute to the increased total moisture export.

Second, we explain why logic “4” is not necessarily correct. The first possibility is that, even if the moisture import from perturbation circulation from the ISM region is nearly zero ($\bar{q} \Delta \nu \sim 0$ out of ISM as shown in Fig.R1b), the total import of moisture from ISM can still be increased by the mean transport $\bar{\nu} \Delta q$ from the ISM. The second possibility can be seen from Fig.R1b: there is still an increase of perturbation circulation of moisture, but now from the western North Pacific, instead of from the ISM. This can be seen from the anomalous circulation $\Delta \nu$ south of NC that originates from the western North Pacific. This suggests that, even though the mean moisture transport is dominated by that from the ISM, there is still the possibility the response of NC rainfall can be enhanced partly by an increase of moisture from the Pacific. Therefore, there is necessarily relation from a stronger ISM to a stronger NC rainfall, even if the mean climatology shows ISM region is the dominant region of moisture source. From another perspective, the correlation of rainfall between ISM and

NC is far from 1. This indicates that they are not perfectly correlated and therefore will not always behave the same, for present day variability and past climate changes.

Fig. R1: Vertical integrated vapor transport from surface to 500 hPa at early Holocene (EH, a), its change from early to mid-Holocene (EH-MH, b) and that from mid- to late Holocene (MH -LH, c). d, South Asia area-averaged rainfall (blue line, 20°-30°N, 85°-95°E) and vertical integrated vapor transport (green line) and zonal wind (red line) from surface to 500 hPa (10°-30°N, 60°-100°E).

2. The relevance to the phase lag

The relation between ISM and NC is not directly relevant to our point of the paper because we are not studying the relation between ISM rainfall and NC rainfall. Indeed, we didn't even specifically study the dynamic cause and moisture source of the rainfall in NC. Our focus is on the phase relation of rainfall and moisture. Nevertheless, it is perhaps useful to include the relation between ISM and NC as part of the mechanism study. Therefore, we have made a further revision.

3. The revision

We have now revised our text to include a further discussion of the relation between ISM and NC rainfall as part of a discussion of the mechanism of NC rainfall response. Since an extensive discussion of this relationship may distract the paper too much, we have tried to keep our discussion brief here. Meanwhile, we have included a new reference, an upcoming paper for the period of deglaciation (He et al., 2021), which has a much more detailed study of the rainfalls in East Asia and ISM, the relation between ISM and NC rainfall, and the explicit study of moisture source and the water isotopes using tagging experiments. (The paper is also attached for the convenience of the reviewers). To further address the question of this reviewer explicitly, we have also added Fig R1 (an enhanced version) to Supplementary Information and added related discussions in the revised text and figure caption.

Our new text on the relation between ISM and NC rainfall are as follows (in Lines 114-126):

“This rainfall decline in NC throughout the Holocene may also be contributed by the enhanced South Asia monsoon via atmospheric teleconnection (23; 31), consistent with the enhanced South Asia monsoon in observations (31-33). The major moisture source to NC appears to originate from the South Asia and Indian Ocean throughout the Holocene (Supplementary Figs. 5a,b,c). This increased rainfall and, then, moisture from the South Asia can be transported by the mean circulation to the NC region and enhance the rainfall there. Furthermore, NC rainfall seems to be also enhanced by the changing circulation and the associated moisture transport from the western North Pacific from the mid- to early Holocene (Supplementary Fig.5d) and from the South Asia and Indian Ocean from late to mid-Holocene (Supplementary Fig.5e). This close relationship between the rainfalls in South Asia and NC, as well as the potential roles of the moisture sources from both the Indian Ocean and the western North Pacific are consistent with a recent study of the Asian monsoon evolution during the last deglaciation (31).”

Ref:

He C., Z. Liu, B. L. Otto-Bliesner, E. C. Brady, C. Zhu, R. Thomas, P. U. Clark, J. Zhu, A. Jahn, S. Gu, J. Zhang, J. Nusbaumer, D. Noone, H. Cheng, Y. Wang, M. Yan and Y. Bao, 2021:

The hydroclimate footprint accompanying pan-Asian monsoon water isotope evolution during the last deglaciation. *Sci. Adv.*, in press. Jan. 22

And on the other hand, new evidence supports that 9.5-5 ka was the wettest time period in NC suggested by stalagmite records of Longfeng cave in NC (Wei et al., 2020 GPC), and stalagmite growth should have no delayed response as ecosystem (as authors suggested). So, the modelling results of 9.5 ka and 6.5 ka (Line 434) could not represent the whole EH (11.7-8.2ka) and MH (8.2-4.2 ka) respectively, as both of the two time-slides are located in the wettest period.

Re: We thank this reviewer for the information of the new moisture record in NC that suggests a peak moisture (in the cave) from 9.5-5ka.

The behavior of this record seems to sit between early Holocene peak (e.g. Supplementary Fig. 4C,D of the paper) and mid-Holocene peak (e.g. Supplementary Fig. 4G,H) and therefore adds an additional valuable data for NC moisture condition. We consider this record neither early Holocene peak, nor mid-Holocene peak. With all these different timings in the different records, we think it indeed even more important to further to understand this hydroclimate evolution of NC in the Holocene. We have included this reference in our revision (in lines 47-48) as “There are also records that peak between the early and mid-Holocene, such as the recent stalagmite records of Longfeng cave in NC (19)”

This record may raise an interesting point on the potentially different evolution of soil moisture at different depths. Our simulation indicated that the soil moisture variations are different between upper and lower layers (not shown but discussed in the content related to Fig. 4). This suggests that the moisture condition of individual cave may differ for different locations, not only because of different rainfall, but also because of different depths. Given the space of this paper, we will leave this interesting question to be studied in the future.

REVIEWERS' COMMENTS

Reviewer #1 (Remarks to the Author):

Thanks for your answers. I have no further comments, but I still keep some my viewpoints. I do suggest that my previous comments (first and last round) and your answers could be published as anonymous reviewer's comments and author's replies, if this MS get published. I believe that there are quite some readers would be interested with the topics we discussed.

Reviewer #1 (Remarks to the Author):

Thanks for your answers. I have no further comments, but I still keep some my viewpoints. I do suggest that my previous comments (first and last round) and your answers could be published as anonymous reviewer's comments and author's replies, if this MS get published. I believe that there are quite some readers would be interested with the topics we discussed.

Re: We thanks reviewer's comments in all round and think they're meaningful to this MS, and we agree to publish the reviewing materials as your suggestions.